# Confidence Intervals and Hypothesis Testing for High-Dimensional Statistical Models

**Adel Javanmard**
Stanford University
Stanford, CA 94305
adelj@stanford.edu

**Andrea Montanari**
Stanford University
Stanford, CA 94305
montanar@stanford.edu

## Abstract

Fitting high-dimensional statistical models often requires the use of non-linear parameter estimation procedures. As a consequence, it is generally impossible to obtain an exact characterization of the probability distribution of the parameter estimates. This in turn implies that it is extremely challenging to quantify the *uncertainty* associated with a certain parameter estimate. Concretely, no commonly accepted procedure exists for computing classical measures of uncertainty and statistical significance as confidence intervals or p-values.

We consider here a broad class of regression problems, and propose an efficient algorithm for constructing confidence intervals and p-values. The resulting confidence intervals have nearly optimal size. When testing for the null hypothesis that a certain parameter is vanishing, our method has nearly optimal power.

Our approach is based on constructing a 'de-biased' version of regularized M-estimators. The new construction improves over recent work in the field in that it does not assume a special structure on the design matrix. Furthermore, proofs are remarkably simple. We test our method on a diabetes prediction problem.

## 1   Introduction

It is widely recognized that modern statistical problems are increasingly high-dimensional, i.e. require estimation of more parameters than the number of observations/examples. Examples abound from signal processing [16], to genomics [21], collaborative filtering [12] and so on. A number of successful estimation techniques have been developed over the last ten years to tackle these problems. A widely applicable approach consists in optimizing a suitably regularized likelihood function. Such estimators are, by necessity, non-linear and non-explicit (they are solution of certain optimization problems).

The use of non-linear parameter estimators comes at a price. In general, it is impossible to characterize the distribution of the estimator. This situation is very different from the one of classical statistics in which either exact characterizations are available, or asymptotically exact ones can be derived from large sample theory [26]. This has an important and very concrete consequence. In classical statistics, generic and well accepted procedures are available for characterizing the uncertainty associated to a certain parameter estimate in terms of confidence intervals or p-values [28, 14]. However, no analogous procedures exist in high-dimensional statistics.

In this paper we develop a computationally efficient procedure for constructing confidence intervals and p-values for a broad class of high-dimensional regression problems. The salient features of our procedure are: $(i)$ Our approach guarantees nearly optimal confidence interval sizes and testing power. $(ii)$ It is the first one that achieves this goal under essentially no assumptions on the population covariance matrix of the parameters, beyond the standard conditions for high-dimensional consistency. $(iii)$ It allows for a streamlined analysis with respect to earlier work in the same area.

**Table 1:** Unbiased estimator for $\theta_0$ in high dimensional linear regression models

---

**Input:** Measurement vector $y$, design matrix $\mathbf{X}$, parameter $\gamma$.

**Output:** Unbiased estimator $\widehat{\theta}^u$.

1: Set $\lambda = \sigma\gamma$, and let $\widehat{\theta}^n$ be the Lasso estimator as per Eq. (3).

2: Set $\widehat{\Sigma} \equiv (\mathbf{X}^\mathsf{T}\mathbf{X})/n$.

3: **for** $i = 1, 2, \ldots, p$ **do**

4:   Let $m_i$ be a solution of the convex program:

$$\begin{aligned} \text{minimize} \quad & m^\mathsf{T}\widehat{\Sigma}m \\ \text{subject to} \quad & \|\widehat{\Sigma}m - e_i\|_\infty \leq \gamma \end{aligned} \tag{4}$$

5: Set $M = (m_1, \ldots, m_p)^\mathsf{T}$. If any of the above problems is not feasible, then set $M = \mathrm{I}_{p\times p}$.

6: Define the estimator $\widehat{\theta}^u$ as follows:

$$\widehat{\theta}^u = \widehat{\theta}^n(\lambda) + \frac{1}{n}\,M\mathbf{X}^\mathsf{T}(Y - \mathbf{X}\widehat{\theta}^n(\lambda)) \tag{5}$$

---

$(iv)$ Our method has a natural generalization non-linear regression models (e.g. logistic regression, see Section 4). We provide heuristic and numerical evidence supporting this generalization, deferring a rigorous study to future work.

For the sake of clarity, we will focus our presentation on the case of linear regression, deferring the generalization to Section 4. In the random design model, we are given $n$ i.i.d. pairs $(Y_1, X_1), (Y_2, X_2), \ldots, (Y_n, X_n)$, with vectors $X_i \in \mathbb{R}^p$ and response variables $Y_i$ given by

$$Y_i = \langle \theta_0, X_i \rangle + W_i, \qquad W_i \sim \mathsf{N}(0, \sigma^2). \tag{1}$$

Here $\langle \cdot, \cdot \rangle$ is the standard scalar product in $\mathbb{R}^p$. In matrix form, letting $Y = (Y_1, \ldots, Y_n)^\mathsf{T}$ and denoting by $\mathbf{X}$ the design matrix with rows $X_1^\mathsf{T}, \ldots, X_n^\mathsf{T}$, we have

$$Y = \mathbf{X}\,\theta_0 + W, \qquad W \sim \mathsf{N}(0, \sigma^2 \mathrm{I}_{n\times n}). \tag{2}$$

The goal is estimate the unknown (but fixed) vector of parameters $\theta_0 \in \mathbb{R}^p$.

In the classic setting, $n \gg p$ and the estimation method of choice is ordinary least squares yielding $\widehat{\theta}^{\mathrm{OLS}} = (\mathbf{X}^\mathsf{T}\mathbf{X})^{-1}\mathbf{X}^\mathsf{T}Y$. In particular $\widehat{\theta}$ is Gaussian with mean $\theta_0$ and covariance $\sigma^2(\mathbf{X}^\mathsf{T}\mathbf{X})^{-1}$. This directly allows to construct confidence intervals[1].

In the high-dimensional setting where $p > n$, the matrix $(\mathbf{X}^\mathsf{T}\mathbf{X})$ is rank deficient and one has to resort to biased estimators. A particularly successful approach is the Lasso [24, 7] which promotes sparse reconstructions through an $\ell_1$ penalty.

$$\widehat{\theta}^n(Y, \mathbf{X}; \lambda) \equiv \arg\min_{\theta \in \mathbb{R}^p} \left\{ \frac{1}{2n}\|Y - \mathbf{X}\theta\|_2^2 + \lambda\|\theta\|_1 \right\}. \tag{3}$$

In case the right hand side has more than one minimizer, one of them can be selected arbitrarily for our purposes. We will often omit the arguments $Y, \mathbf{X}$, as they are clear from the context. We denote by $S \equiv \mathrm{supp}(\theta_0) \subseteq [p]$ the support of $\theta_0$, and let $s_0 \equiv |S|$. A copious theoretical literature [6, 2, 4] shows that, under suitable assumptions on $\mathbf{X}$, the Lasso is nearly as accurate as if the support $S$ was known *a priori*. Namely, for $n = \Omega(s_0 \log p)$, we have $\|\widehat{\theta}^n - \theta_0\|_2^2 = O(s_0\sigma^2(\log p)/n)$. These remarkable properties come at a price. Deriving an exact characterization for the distribution of $\widehat{\theta}^n$ is not tractable in general, and hence there is no simple procedure to construct confidence intervals and p-values. In order to overcome this challenge, we construct a de-biased estimator from the Lasso solution. The de-biased estimator is given by the simple formula $\widehat{\theta}^u = \widehat{\theta}^n + (1/n)\,M\mathbf{X}^\mathsf{T}(Y - \mathbf{X}\widehat{\theta}^n)$, as in Eq. (5). The basic intuition is that $\mathbf{X}^\mathsf{T}(Y - \mathbf{X}\widehat{\theta}^n)/(n\lambda)$ is a subgradient of the $\ell_1$ norm at the Lasso solution $\widehat{\theta}_n$. By adding a term proportional to this subgradient, our procedure compensates the bias introduced by the $\ell_1$ penalty in the Lasso.

We will prove in Section 2 that $\widehat{\theta}^u$ is approximately Gaussian, with mean $\theta_0$ and covariance $\sigma^2(M\widehat{\Sigma}M)/n$, where $\widehat{\Sigma} = (\mathbf{X}^\mathsf{T}\mathbf{X}/n)$ is the empirical covariance of the feature vectors. This result allows to construct confidence intervals and p-values in complete analogy with classical statistics procedures. For instance, letting $Q \equiv M\widehat{\Sigma}M$, $[\widehat{\theta}_i^u - 1.96\sigma\sqrt{Q_{ii}/n}, \widehat{\theta}_i^u + 1.96\sigma\sqrt{Q_{ii}/n}]$ is a 95% confidence interval. The size of this interval is of order $\sigma/\sqrt{n}$, which is the optimal (minimum) one, i.e. the same that would have been obtained by knowing *a priori* the support of $\theta_0$. In practice the noise standard deviation is not known, but $\sigma$ can be replaced by any consistent estimator $\widehat{\sigma}$.

A key role is played by the matrix $M \in \mathbb{R}^{p \times p}$ whose function is to 'decorrelate' the columns of $\mathbf{X}$. We propose here to construct $M$ by solving a convex program that aims at optimizing two objectives. One one hand, we try to control $|M\widehat{\Sigma} - \mathrm{I}|_\infty$ (here and below $|\cdot|_\infty$ denotes the entrywise $\ell_\infty$ norm) which –as shown in Theorem 2.1– controls the non-Gaussianity and bias of $\widehat{\theta}^u$. On the other, we minimize $[M\widehat{\Sigma}M]_{i,i}$, for each $i \in [p]$, which controls the variance of $\widehat{\theta}_i^u$.

The idea of constructing a de-biased estimator of the form $\widehat{\theta}^u = \widehat{\theta}^n + (1/n)\,M\mathbf{X}^\mathsf{T}(Y - \mathbf{X}\widehat{\theta}^n)$ was used by Javanmard and Montanari in [10], that suggested the choice $M = c\Sigma^{-1}$, with $\Sigma = \mathbb{E}\{X_1 X_1^\mathsf{T}\}$ the population covariance matrix and $c$ a positive constant. A simple estimator for $\Sigma$ was proposed for sparse covariances, but asymptotic validity and optimality were proven only for uncorrelated Gaussian designs (i.e. Gaussian $\mathbf{X}$ with $\Sigma = \mathrm{I}$). Van de Geer, Bülhmann and Ritov [25] used the same construction with $M$ an estimate of $\Sigma^{-1}$ which is appropriate for sparse inverse covariances. These authors prove semi-parametric optimality in a non-asymptotic setting, provided the sample size is at least $n = \Omega(s_0^2 \log p)$. In this paper, we do not assume any sparsity constraint on $\Sigma^{-1}$, but still require the sample size scaling $n = \Omega(s_0^2 \log p)$. We refer to a forthcoming publication wherein the condition on the sample size scaling is relaxed [11].

From a technical point of view, our proof starts from a simple decomposition of the de-biased estimator $\widehat{\theta}^u$ into a Gaussian part and an error term, already used in [25]. However –departing radically from earlier work– we realize that $M$ need not be a good estimator of $\Sigma^{-1}$ in order for the de-biasing procedure to work. We instead set $M$ as to minimize the error term and the variance of the Gaussian term. As a consequence of this choice, our approach applies to general covariance structures $\Sigma$. By contrast, earlier approaches applied only to sparse $\Sigma$, as in [10], or sparse $\Sigma^{-1}$ as in [25]. The only assumptions we make on $\Sigma$ are the standard compatibility conditions required for high-dimensional consistency [4]. We refer the reader to the long version of the paper [9] for the proofs of our main results and the technical steps.

## 1.1 Further related work

The theoretical literature on high-dimensional statistical models is vast and rapidly growing. Restricting ourselves to linear regression, earlier work investigated prediction error [8], model selection properties [17, 31, 27, 5], $\ell_2$ consistency [6, 2]. Of necessity, we do not provide a complete set of references, and instead refer the reader to [4] for an in-depth introduction to this area.

The problem of quantifying statistical significance in high-dimensional parameter estimation is, by comparison, far less understood. Zhang and Zhang [30], and Bühlmann [3] proposed hypothesis testing procedures under restricted eigenvalue or compatibility conditions [4]. These methods are however effective only for detecting very large coefficients. Namely, they both require $|\theta_{0,i}| \geq c \max\{\sigma s_0 \log p/\,n, \sigma/\sqrt{n}\}$, which is $\sqrt{s_0}$ larger than the ideal detection level [10]. In other words, in order for the coefficient $\theta_{0,i}$ to be detectable with appreciable probability, it needs to be larger than the overall $\ell_2$ error, rather than the $\ell_2$ error per coordinate.

Lockart et al. [15] develop a test for the hypothesis that a newly added coefficient along the Lasso regularization path is irrelevant. This however does not allow to test arbitrary coefficients at a given value of $\lambda$, which is instead the problem addressed in this paper. It further assumes that the current Lasso support contains the actual support $\mathrm{supp}(\theta_0)$ and that the latter has bounded size. Finally, resampling methods for hypothesis testing were studied in [29, 18, 19].

## 1.2 Preliminaries and notations

We let $\widehat{\Sigma} \equiv \mathbf{X}^\mathsf{T}\mathbf{X}/n$ be the sample covariance matrix. For $p > n$, $\widehat{\Sigma}$ is always singular. However, we may require $\widehat{\Sigma}$ to be nonsingular for a restricted set of directions.

**Definition 1.1.** *For a matrix $\widehat{\Sigma}$ and a set $S$ of size $s_0$, the* compatibility condition *is met, if for some $\phi_0 > 0$, and all $\theta$ satisfying $\|\theta_{S^c}\|_1 \leq 3\|\theta_S\|_1$, it holds that*

$$\|\theta_S\|_1^2 \leq \frac{s_0}{\phi_0^2} \theta^\mathsf{T} \widehat{\Sigma} \theta \,.$$

**Definition 1.2.** *The* sub-gaussian norm *of a random variable $X$, denoted by $\|X\|_{\psi_2}$, is defined as*

$$\|X\|_{\psi_2} = \sup_{p \geq 1} p^{-1/2} (\mathbb{E}|X|^p)^{1/p} \,.$$

*The sub-gaussian norm of a random vector $X \in \mathbb{R}^n$ is defined as $\|X\|_{\psi_2} = \sup_{x \in S^{n-1}} \|\langle X, x \rangle\|_{\psi_2}$. Further, for a random variable $X$, its* sub-exponential norm*, denoted by $\|X\|_{\psi_1}$, is defined as*

$$\|X\|_{\psi_1} = \sup_{p \geq 1} p^{-1} (\mathbb{E}|X|^p)^{1/p} \,.$$

For a matrix $A$ and set of indices $I, J$, we let $A_{I,J}$ denote the submatrix formed by the rows in $I$ and columns in $J$. Also, $A_{I,\cdot}$ (resp. $A_{\cdot,I}$) denotes the submatrix containing just the rows (reps. columns) in $I$. Likewise, for a vector $v$, $v_I$ is the restriction of $v$ to indices in $I$. We use the shorthand $A_{I,J}^{-1} = (A^{-1})_{I,J}$. In particular, $A_{i,i}^{-1} = (A^{-1})_{i,i}$. The maximum and the minimum singular values of $A$ are respectively denoted by $\sigma_{\max}(A)$ and $\sigma_{\min}(A)$. We write $\|v\|_p$ for the standard $\ell_p$ norm of a vector $v$ and $\|v\|_0$ for the number of nonzero entries of $v$. For a matrix $A$, $\|A\|_p$ is the $\ell_p$ operator norm, and $|A|_p$ is the elementwise $\ell_p$ norm, i.e., $|A|_p = (\sum_{i,j} |A_{ij}|^p)^{1/p}$. For an integer $p \geq 1$, we let $[p] \equiv \{1, \ldots, p\}$. For a vector $v$, $\mathrm{supp}(v)$ represents the positions of nonzero entries of $v$. Throughout, with high probability (w.h.p) means with probability converging to one as $n \to \infty$, and $\Phi(x) \equiv \int_{-\infty}^x e^{-t^2/2} \mathrm{d}t / \sqrt{2\pi}$ denotes the CDF of the standard normal distribution.

## 2 An de-biased estimator for $\theta_0$

**Theorem 2.1.** *Consider the linear model* (1) *and let $\widehat{\theta}^u$ be defined as per Eq.* (5). *Then,*

$$\sqrt{n}(\widehat{\theta}^u - \theta_0) = Z + \Delta \,, \quad Z|\mathbf{X} \sim \mathsf{N}(0, \sigma^2 M\widehat{\Sigma}M^\mathsf{T}) \,, \quad \Delta = \sqrt{n}(M\widehat{\Sigma} - \mathrm{I})(\theta_0 - \widehat{\theta}) \,.$$

*Further, suppose that $\sigma_{\min}(\Sigma) = \Omega(1)$, and $\sigma_{\max}(\Sigma) = O(1)$. In addition assume the rows of the whitened matrix $\mathbf{X}\Sigma^{-1/2}$ are sub-gaussian, i.e., $\|\Sigma^{-1/2}X_1\|_{\psi_2} = O(1)$. Let $\mathcal{E}$ be the event that the compatibility condition holds for $\widehat{\Sigma}$, and $\max_{i \in [p]} \widehat{\Sigma}_{i,i} = O(1)$. Then, using $\gamma = O(\sqrt{(\log p)/n})$ (see inputs in Table 1), the following holds true. On the event $\mathcal{E}$, w.h.p, $\|\Delta\|_\infty = O(s_0 \log p / \sqrt{n})$.*

Note that compatibility condition (and hence the event $\mathcal{E}$) holds w.h.p. for random design matrices of a general nature. In fact [22] shows that under some general assumptions, the compatibility condition on $\Sigma$ implies a similar condition on $\widehat{\Sigma}$, w.h.p., when $n$ is sufficiently large. Bounds on the variances $[M\widehat{\Sigma}M^\mathsf{T}]_{ii}$ will be given in Section 3.2. Finally, the claim of Theorem 2.1 does not rely on the specific choice of the objective function in optimization problem (4) and only uses the optimization constraints.

**Remark 2.2.** *Theorem 2.1 does not make any assumption about the parameter vector $\theta_0$. If we further assume that the support size $s_0$ satisfies $s_0 = o(\sqrt{n}/\log p)$, then we have $\|\Delta\|_\infty = o(1)$, w.h.p. Hence, $\widehat{\theta}^u$ is an asymptotically unbiased estimator for $\theta_0$.*

## 3 Statistical inference

A direct application of Theorem 2.1 is to derive confidence intervals and statistical hypothesis tests for high dimensional models. Throughout, we make the sparsity assumption $s_0 = o(\sqrt{n}/\log p)$.

### 3.1 Confidence intervals

We first show that the variances of variables $Z_j|\mathbf{X}$ are $\Omega(1)$.

**Lemma 3.1.** *Let $M = (m_1, \ldots, m_p)^{\mathsf{T}}$ be the matrix with rows $m_i^{\mathsf{T}}$ obtained by solving convex program* (4). *Then for all $i \in [p]$, $[M\widehat{\Sigma}M^{\mathsf{T}}]_{i,i} \geq (1-\gamma)^2/\widehat{\Sigma}_{i,i}$.*

By Remark 2.2 and Lemma 3.1, we have

$$\mathbb{P}\left\{ \frac{\sqrt{n}(\widehat{\theta}_i^u - \theta_{0,i})}{\sigma[M\widehat{\Sigma}M^{\mathsf{T}}]_{i,i}^{1/2}} \leq x \Big| \mathbf{X} \right\} = \Phi(x) + o(1), \qquad \forall x \in \mathbb{R}. \tag{6}$$

Since the limiting probability is independent of $\mathbf{X}$, Eq. (6) also holds unconditionally for random design $\mathbf{X}$.

For constructing confidence intervals, a consistent estimate of $\sigma$ is needed. To this end, we use the scaled Lasso [23] given by

$$\{\widehat{\theta}^n(\lambda), \widehat{\sigma}\} \equiv \underset{\theta \in \mathbb{R}^p, \sigma > 0}{\arg\min} \left\{ \frac{1}{2\sigma n}\|Y - \mathbf{X}\theta\|_2^2 + \frac{\sigma}{2} + \lambda\|\theta\|_1 \right\}.$$

This is a joint convex minimization which provides an estimate of the noise level in addition to an estimate of $\theta_0$. We use $\lambda = c_1\sqrt{(\log p)/n}$ that yields a consistent estimate $\widehat{\sigma}$, under the assumptions of Theorem 2.1 (cf. [23]). We hence obtain the following.

**Corollary 3.2.** *Let*

$$\delta(\alpha, n) = \Phi^{-1}(1 - \alpha/2)\widehat{\sigma}\, n^{-1/2}\sqrt{[M\widehat{\Sigma}M^{\mathsf{T}}]_{i,i}}. \tag{7}$$

*Then $I_i = [\widehat{\theta}_i^u - \delta(\alpha, n), \widehat{\theta}_i^u + \delta(\alpha, n)]$ is an asymptotic two-sided confidence interval for $\theta_{0,i}$ with significance $\alpha$.*

Notice that the same corollary applies to any other consistent estimator $\widehat{\sigma}$ of the noise standard deviation.

## 3.2 Hypothesis testing

An important advantage of sparse linear regression models is that they provide parsimonious explanations of the data in terms of a small number of covariates. The easiest way to select the 'active' covariates is to choose the indexes $i$ for which $\widehat{\theta}_i^n \neq 0$. This approach however does not provide a measure of statistical significance for the finding that the coefficient is non-zero.

More precisely, we are interested in testing an individual null hypothesis $H_{0,i} : \theta_{0,i} = 0$ versus the alternative $H_{A,i} : \theta_{0,i} \neq 0$, and assigning $p$-values for these tests. We construct a $p$-value $P_i$ for the test $H_{0,i}$ as follows:

$$P_i = 2\left(1 - \Phi\left(\frac{\sqrt{n}\,|\widehat{\theta}_i^u|}{\widehat{\sigma}[M\widehat{\Sigma}M^{\mathsf{T}}]_{i,i}^{1/2}}\right)\right). \tag{8}$$

The decision rule is then based on the $p$-value $P_i$:

$$T_{i,\mathbf{X}}(y) = \begin{cases} 1 & \text{if } P_i \leq \alpha \qquad (\text{reject } H_{0,i}), \\ 0 & \text{otherwise} \qquad (\text{accept } H_{0,i}). \end{cases} \tag{9}$$

We measure the quality of the test $T_{i,\mathbf{X}}(y)$ in terms of its significance level $\alpha_i$ and statistical power $1 - \beta_i$. Here $\alpha_i$ is the probability of type I error (i.e. of a false positive at $i$) and $\beta_i$ is the probability of type II error (i.e. of a false negative at $i$).

Note that it is important to consider the tradeoff between statistical significance and power. Indeed any significance level $\alpha$ can be achieved by randomly rejecting $H_{0,i}$ with probability $\alpha$. This test achieves power $1 - \beta = \alpha$. Further note that, without further assumption, no nontrivial power can be achieved. In fact, choosing $\theta_{0,i} \neq 0$ arbitrarily close to zero, $H_{0,i}$ becomes indistinguishable from its alternative. We will therefore assume that, whenever $\theta_{0,i} \neq 0$, we have $|\theta_{0,i}| > \mu$ as well. We take a minimax perspective and require the test to behave uniformly well over $s_0$-sparse vectors. Formally, for $\mu > 0$ and $i \in [p]$, define

$$\alpha_i(n) \equiv \sup\left\{ \mathbb{P}_{\theta_0}(T_{i,\mathbf{X}}(y) = 1) : \ \theta_0 \in \mathbb{R}^p, \ \|\theta_0\|_0 \leq s_0(n), \ \theta_{0,i} = 0 \right\}.$$

$$\beta_i(n; \mu) \equiv \sup\left\{ \mathbb{P}_{\theta_0}(T_{i,\mathbf{X}}(y) = 0) : \ \theta_0 \in \mathbb{R}^p, \ \|\theta_0\|_0 \leq s_0(n), \ |\theta_{0,i}| \geq \mu \right\}.$$

Here, we made dependence on $n$ explicit. Also, $\mathbb{P}_\theta(\cdot)$ is the induced probability for random design $\mathbf{X}$ and noise realization $w$, given the fixed parameter vector $\theta$. Our next theorem establishes bounds on $\alpha_i(n)$ and $\beta_i(n; \mu)$.

**Theorem 3.3.** *Consider a random design model that satisfies the conditions of Theorem 2.1. Under the sparsity assumption $s_0 = o(\sqrt{n}/\log p)$, the following holds true for any fixed sequence of integers $i = i(n)$:*

$$\lim_{n \to \infty} \alpha_i(n) \leq \alpha. \tag{10}$$

$$\lim_{n \to \infty} \frac{1 - \beta_i(\mu; n)}{1 - \beta_i^*(\mu; n)} \geq 1, \qquad 1 - \beta_i^*(\mu; n) \equiv G\left(\alpha, \frac{\sqrt{n}\,\mu}{\sigma[\Sigma_{i,i}^{-1}]^{1/2}}\right), \tag{11}$$

*where, for $\alpha \in [0, 1]$ and $u \in \mathbb{R}_+$, the function $G(\alpha, u)$ is defined as follows:*

$$G(\alpha, u) = 2 - \Phi(\Phi^{-1}(1 - \frac{\alpha}{2}) + u) - \Phi(\Phi^{-1}(1 - \frac{\alpha}{2}) - u).$$

It is easy to see that, for any $\alpha > 0$, $u \mapsto G(\alpha, u)$ is continuous and monotone increasing. Moreover, $G(\alpha, 0) = \alpha$ which is the trivial power obtained by randomly rejecting $H_{0,i}$ with probability $\alpha$. As $\mu$ deviates from zero, we obtain nontrivial power. Notice that in order to achieve a specific power $\beta > \alpha$, our scheme requires $\mu = O(\sigma/\sqrt{n})$, since $\Sigma_{i,i}^{-1} \leq \sigma_{\max}(\Sigma^{-1}) \leq (\sigma_{\min}(\Sigma))^{-1} = O(1)$.

### 3.2.1 Minimax optimality

The authors of [10] prove an upper bound for the minimax power of tests with a given significance level $\alpha$, under the Gaussian random design models (see Theorem 2.6 therein). This bound is obtained by considering an oracle test that knows all the active parameters except $i$, i.e., $S \backslash \{i\}$. To state the bound formally, for a set $S \subseteq [p]$ and $i \in S^c$, define $\Sigma_{i|S} \equiv \Sigma_{i,i} - \Sigma_{i,S}(\Sigma_{S,S})^{-1}\Sigma_{S,i}$, and let

$$\eta_{\Sigma, s_0} \equiv \min_{i \in [p], S} \left\{ \Sigma_{i|S} : \ S \subseteq [p] \backslash \{i\}, |S| < s_0 \right\}.$$

In asymptotic regime and under our sparsity assumption $s_0 = o(\sqrt{n}/\log p)$, the bound of [10] simplifies to

$$\lim_{n \to \infty} \frac{1 - \beta_i^{\mathrm{opt}}(\alpha; \mu)}{G(\alpha, \mu/\sigma_{\mathrm{eff}})} \leq 1, \quad \sigma_{\mathrm{eff}} = \frac{\sigma}{\sqrt{n\,\eta_{\Sigma, s_0}}}, \tag{12}$$

Using the bound of (12) and specializing the result of Theorem 3.3 to Gaussian design $\mathbf{X}$, we obtain that our scheme achieves a near optimal minimax power for a broad class of covariance matrices. We can compare our test to the optimal test by computing how much $\mu$ must be increased in order to achieve the minimax optimal power. It follows from the above that $\mu$ must be increased to $\tilde{\mu}$, with the two differing by a factor:

$$\tilde{\mu}/\mu = \sqrt{\Sigma_{ii}^{-1}\,\eta_{\Sigma, s_0}} \leq \sqrt{\Sigma_{i,i}^{-1}\Sigma_{i,i}} \leq \sqrt{\sigma_{\max}(\Sigma)/\sigma_{\min}(\Sigma)},$$

since $\Sigma_{ii}^{-1} \leq (\sigma_{\min}(\Sigma))^{-1}$, and $\Sigma_{i|S} \leq \Sigma_{i,i} \leq \sigma_{\max}(\Sigma)$ due to $\Sigma_{S,S} \succ 0$.

## 4 General regularized maximum likelihood

In this section, we generalize our results beyond the linear regression model to general regularized maximum likelihood. Here, we only describe the de-biasing method. Formal guarantees can be obtained under suitable restricted strong convexity assumptions [20] and will be the object of a forthcoming publication.

For univariate $Y$, and vector $X \in \mathbb{R}^p$, we let $\{f_\theta(Y|X)\}_{\theta \in \mathbb{R}^p}$ be a family of conditional probability densities parameterized by $\theta$, that are absolutely continuous with respect to a common measure $\omega(dy)$, and suppose that the gradient $\nabla_\theta f_\theta(Y|X)$ exists and is square integrable.

As in for linear regression, we assume that the data is given by $n$ i.i.d. pairs $(X_1, Y_1), \ldots (X_n, Y_n)$, where conditional on $X_i$, the response variable $Y_i$ is distributed as

$$Y_i \sim f_{\theta_0}(\cdot | X_i).$$

for some parameter vector $\theta_0 \in \mathbb{R}^p$. Let $\mathcal{L}_i(\theta) = -\log f_\theta(Y_i|X_i)$ be the normalized negative log-likelihood corresponding to the observed pair $(Y_i, X_i)$, and define $\mathcal{L}(\theta) = \frac{1}{n}\sum_{i=1}^n \mathcal{L}_i(\theta)$. We consider the following regularized estimator:

$$\widehat{\theta} \equiv \arg\min_{\theta\in\mathbb{R}^p} \left\{ \mathcal{L}(\theta) + \lambda\mathcal{R}(\theta) \right\}, \tag{13}$$

where $\lambda$ is a regularization parameter and $\mathcal{R} : \mathbb{R}^p \to \mathbb{R}_+$ is a norm.

We next generalize the definition of $\widehat{\Sigma}$. Let $\mathcal{I}_i(\theta)$ be the Fisher information of $f_\theta(Y|X_i)$, defined as

$$\mathcal{I}_i(\theta) \equiv \mathbb{E}\left[\left(\nabla_\theta \log f_\theta(Y|X_i)\right)\left(\nabla_\theta \log f_\theta(Y|X_i)\right)^\mathsf{T}\Big|X_i\right] = -\mathbb{E}\left[\left(\nabla_\theta^2 \log f(Y|X_i,\theta)\right)\Big|X_i\right],$$

where the second identity holds under suitable regularity conditions [13], and $\nabla_\theta^2$ denotes the Hessian operator. We assume $\mathbb{E}[\mathcal{I}_i(\theta)] \succ 0$ define $\widehat{\Sigma} \in \mathbb{R}^{p\times p}$ as follows:

$$\widehat{\Sigma} \equiv \frac{1}{n}\sum_{i=1}^n \mathcal{I}_i(\widehat{\theta}). \tag{14}$$

Note that (in general) $\widehat{\Sigma}$ depends on $\widehat{\theta}$. Finally, the de-biased estimator $\widehat{\theta}^u$ is defined by $\widehat{\theta}^u \equiv \widehat{\theta} - M\nabla_\theta\mathcal{L}(\widehat{\theta})$, with $M$ given again by the solution of the convex program (4), and the definition of $\widehat{\Sigma}$ provided here. Notice that this construction is analogous to the one in [25] (although the present setting is somewhat more general) with the crucial difference of the construction of $M$.

A a simple heuristic derivation of this method is the following. By Taylor expansion of $\mathcal{L}(\widehat{\theta})$ around $\theta_0$ we get $\widehat{\theta}^u \approx \widehat{\theta} - M\nabla_\theta\mathcal{L}(\theta_0) - M\nabla_\theta^2\mathcal{L}(\theta_0)(\widehat{\theta}-\theta_0)$. Approximating $\nabla_\theta^2\mathcal{L}(\theta_0) \approx \widehat{\Sigma}$ (which amounts to taking expectation with respect to the response variables $y_i$), we get $\widehat{\theta}^u - \theta_0 \approx -M\nabla_\theta\mathcal{L}(\theta_0) - [M\widehat{\Sigma} - \mathrm{I}](\widehat{\theta}-\theta_0)$. Conditionally on $\{X_i\}_{1\leq i\leq n}$, the first term has zero expectation and covariance $[M\widehat{\Sigma}M]$. Further, by central limit theorem, its low-dimensional marginals are approximately Gaussian. The bias term $-[M\widehat{\Sigma} - \mathrm{I}](\widehat{\theta}-\widehat{\theta}_0)$ can be bounded as in the linear regression case, building on the fact that $M$ is chosen such that $|M\widehat{\Sigma} - \mathrm{I}|_\infty \leq \gamma$.

Similar to the linear case, an asymptotic two-sided confidence interval for $\theta_{0,i}$ (with significance $\alpha$) is given by $I_i = [\widehat{\theta}_i^u - \delta(\alpha, n), \widehat{\theta}_i^u + \delta(\alpha, n)]$, where

$$\delta(\alpha, n) = \Phi^{-1}(1 - \alpha/2)n^{-1/2}[M\widehat{\Sigma}M^\mathsf{T}]_{i,i}^{1/2}.$$

Moreover, an asymptotically valid p-value $P_i$ for testing null hypothesis $H_{0,i}$ is constructed as:

$$P_i = 2\left(1 - \Phi\left(\frac{\sqrt{n}|\widehat{\theta}_i^u|}{[M\widehat{\Sigma}M^\mathsf{T}]_{i,i}^{1/2}}\right)\right).$$

In the next section, we shall apply the general approach presented here to $\mathcal{L}_1$-regularized logistic regression. In this case, the binary response $Y_i \in \{0, 1\}$ is distributed as $Y_i \sim f_{\theta_0}(\cdot|X_i)$ where $f_{\theta_0}(1|x) = (1 + e^{-\langle x,\theta_0\rangle})^{-1}$ and $f_{\theta_0}(0|x) = (1 + e^{\langle x,\theta_0\rangle})^{-1}$. It is easy to see that in this case $\mathcal{I}_i(\widehat{\theta}) = \widehat{q}_i(1 - \widehat{q}_i)X_iX_i^\mathsf{T}$, with $\widehat{q}_i = (1 + e^{-\langle\widehat{\theta},X_i\rangle})^{-1}$, and thus

$$\widehat{\Sigma} = \frac{1}{n}\sum_{i=1}^n \widehat{q}_i(1 - \widehat{q}_i)X_iX_i^\mathsf{T}.$$

## 5  Diabetes data example

We consider the problem of estimating relevant attributes in predicting type-2 diabetes. We evaluate the performance of our hypothesis testing procedure on the Practice Fusion Diabetes dataset [1]. This dataset contains de-identified medical records of 10000 patients, including information on diagnoses, medications, lab results, allergies, immunizations, and vital signs. From this dataset, we extract $p$ numerical attributes resulting in a sparse design matrix $\mathbf{X}_{\text{tot}} \in \mathbb{R}^{n_{\text{tot}}\times p}$, with $n_{\text{tot}} = 10000$,

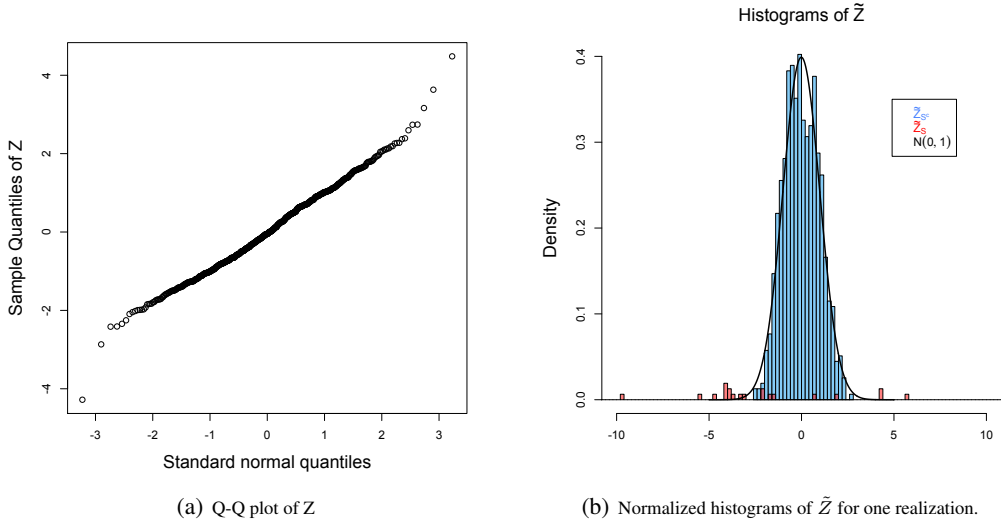

(a) Q-Q plot of Z

(b) Normalized histograms of $\tilde{Z}$ for one realization.

Figure 1: Q-Q plot of $Z$ and normalized histograms of $\tilde{Z}_S$ (in red) and $\tilde{Z}_{S^c}$ (in blue) for one realization. No fitting of the Gaussian mean and variance was done in panel (b).

and $p = 805$ (only $5.9\%$ entries of $\mathbf{X}_{\text{tot}}$ are non-zero). Next, we standardize the columns of $\mathbf{X}$ to have mean $0$ and variance $1$. The attributes consist of: $(i)$*Transcript records:* year of birth, gender and BMI; $(ii)$*Diagnoses informations:* 80 binary attributes corresponding to different ICD-9 codes. $(iii)$*Medications:* 80 binary attributes indicating the use of different medications. $(iv)$ *Lab results:* For 70 lab test observations, we include attributes indicating patients tested, abnormality flags, and the observed values. We also bin the observed values into 10 quantiles and make 10 binary attributes indicating the bin of the corresponding observed value.

We consider logistic model as described in the previous section with a binary response identifying the patients diagnosed with type-2 diabetes. For the sake of performance evaluation, we need to know the true significant attributes. Letting $\mathcal{L}(\theta)$ be the logistic loss corresponding to the design $\mathbf{X}_{\text{tot}}$ and response vector $Y \in \mathbb{R}^{n_{\text{tot}}}$, we take $\theta_0$ as the minimizer of $\mathcal{L}(\theta)$. Notice that here, we are in the low dimensional regime ($n_{\text{tot}} > p$) and no regularization is needed.

Next, we take random subsamples of size $n = 500$ from the patients, and examine the performance of our testing procedure. The experiment is done using glmnet-package in R that fits the entire path of the regularized logistic estimator. We then choose the value of $\lambda$ that yields maximum AUC (area under ROC curve), approximated by a 5-fold cross validation.

*Results*: Type I errors and powers of our decision rule (9) are computed by comparing to $\theta_0$. The average error and power (over 20 random subsamples) and significance level $\alpha = 0.05$ are respectively, 0.0319 and 0.818. Let $Z = (z_i)_{i=1}^p$ denote the vector with $z_i \equiv \sqrt{n}(\widehat{\theta}_i^u - \theta_{0,i})/[M\widehat{\Sigma}M]_{i,i}^{1/2}$. In Fig. 1(a), sample quantiles of $Z$ are depicted versus the quantiles of a standard normal distribution. The plot clearly corroborates our theoretical result regarding the limiting distribution of $Z$. In order to build further intuition about the proposed $p$-values, let $\tilde{Z} = (\tilde{z}_i)_{i=1}^p$ be the vector with $\tilde{z}_i \equiv \sqrt{n}\widehat{\theta}_i^u/[M\widehat{\Sigma}M]_{i,i}^{1/2}$. In Fig. 1(b), we plot the normalized histograms of $\tilde{Z}_S$ (in red) and $\tilde{Z}_{S^c}$ (in blue). As the plot showcases, $\tilde{Z}_{S^c}$ has roughly standard normal distribution, and the entries of $\tilde{Z}_S$ appear as distinguishable spikes. The entries of $\tilde{Z}_S$ with larger magnitudes are easier to be marked off from the normal distribution tail.

## Footnotes

[1] For instance, letting $Q \equiv (\mathbf{X}^\mathsf{T}\mathbf{X}/n)^{-1}$, $\widehat{\theta}_i^{\mathrm{OLS}} - 1.96\sigma\sqrt{Q_{ii}/n}, \widehat{\theta}_i^{\mathrm{OLS}} + 1.96\sigma\sqrt{Q_{ii}/n}]$ is a 95% confidence interval [28].

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
