[Reviews · NeurIPS 2013]

Submitted by Assigned_Reviewer_1

The paper improves on the work in [1], which in turn is related to [2]. The improvement is in the conditions that are needed for establishing confidence bounds.

The difference to [1] comes from the way that Sigma^{-1} is estimated. While [1] uses neighborhood selection procedure [3] to invert Sigma, in this paper Sigma is approximately inverted using a constrained QP.

The paper is technically interesting and correct.

Have two questions:

1) In order to obtain confidence intervals you still require that s = o(sqrt(n)/log(p)). See Remark 2.2. This is needed for the term ||Delta||_inf to be o(1). However, this seems to be a more stringent requirement than what we normally need for point estimation. That is, you require n = Omega(s^2 log(p)^2) instead of n = Omega(s log(p)). Is this correct or am I missing something? If that is correct, maybe you should mention in intro that you require a bit more stringent assumptions than what is needed for estimation.

2) I strongly encourage you to run simulations studies that demonstrate coverage of your intervals. Do your confidence intervals correctly capture the true parameter 95%, say, of times? What about size and power of the test? I think this would be more insightful than an illustration on diabetes data. Also, you should compare against the procedure in [2]. How different are the confidence intervals using these two procedures?

[1] S. van de Geer, P. Buhlmann, and Y. Ritov. On asymptotically optimal confidence regions and tests for high-dimensional models. arXiv:1303.0518, 2013.

[2] Cun-Hui Zhang, Stephanie S. Zhang. Confidence Intervals for Low-Dimensional Parameters in High-Dimensional Linear Models. arXiv:1110.2563

[3] N. Meinshausen and P. Buhlmann. High-dimensional graphs and variable selection with the Lasso. Annals of Statistics, 34:1436–1462, 2006.
Summary: The paper deals with an important problem of providing confidence intervals to parameters in high-dimensional regression. It builds on the recent work by van de Geer, Buhlmann and Ritov. Simulation results would be very helpful.

Submitted by Assigned_Reviewer_4

The paper constructs confidence intervals for the elements of the parameter vector in sparse linear regression. The results hold when the inputs are sub-Gaussian, the noise is Gaussian, and the empirical covariance matrix satisfies a compatibility condition.

The paper constructs a "de-biased" estimator t1 from the LASSO estimate and shows that t1 is approximately Gaussian with a data-dependent covariance matrix and mean t0, where t0 is the true parameter vector. Given this result, confidence intervals of the form [t1_i-C/sqrt{n}, t1_i+C/sqrt{n}] are obtained, where n is the sample size and C is a constant that does not depend on the sparsity level s.

The main claim of the paper is a confidence interval that does not scale with sparsity. Authors mention paper [23] that obtains similar results but requires n = Omega(s^2 log( p )). But isn't this also required in the current paper? In Section 3, authors make the "sparsity assumption" that s = o(sort{n}/log p), which implies that n = Omega(s^2 log^2( p )). What is the improvement compared to [23]? Theorem 2.1 in the current paper is very similar to Theorems 2.1 and 2.2 in [23].

Now back to the main result of the paper, it is not clear that the result holds. Theorem 2.1 is stated in an asymptotic form and the non-asymptotic version is shown in Appendix A. But the result in Appendix A only states that Delta is small with some constant probability and this probability is not getting bigger with more samples. How do you get Delta = o(1) from this?

Other issues:

Theorem 2.1 assumes that the compatibility condition holds for hat{Sigma}. Then it is said that this holds w.h.p if the compatibility condition holds for Sigma and some other conditions are satisfied. Maybe it is better to explicitly state all these conditions under which Thm 2.1 holds.

================================

After author feedback:

Thanks for the feedback. Please add these explanations to the paper. I will change my score.
Summary: The paper constructs confidence intervals for the elements of the parameter vector in sparse linear regression. The problem is interesting and worthwhile. Overall, the paper is well-written, although some details should be added to make the distinctions clearer.

Submitted by Assigned_Reviewer_5

CONTENT

The paper proposes a new method to constructing confidence
intervals (CI's) and to perform testing of individual parameters
in high-dimensional problems of supervised learning under
the sparsity assumption. Although the main focus is on the
model of linear regression, an extension to more general
cases is presented. A numerical illustration of the proposed
methodology is carried out on the diabetes data.

The main observation underlying this work is that in the
high-dimensional Gaussian linear model, the LSE is inconsistent
but unbiased and has a Gaussian distribution with a well-known
covariance matrix. Thus it can be used for constructing CI's.
On the other hand, the Lasso is consistent under suitable
assumptions but has a relatively large bias and, therefore, cannot
be used for constructing CI's. The solution, that has been already
evoked in previous work by Javanmard and Montanari [9] and
van de Geer et al. [23], is to construct CI's based on an estimator
which is between the LSE and the Lasso, which has a bias that is
small as compared to the variance, under suitable assumptions.

The most important question here is which assumptions may be considered
as ``acceptable'' in this context. Basically, one can simply use
the LSE \hat\theta = pinv(X^T*X)*X^T*Y, where pinv(M) is the
pseudo-inverse of a matrix M, and construct a CI based on it. The
length of the corresponding CI for the j-th parameter will be
\sigma*sqrt(pinv(X^T*X)_{jj}). Under some (very strong) assumptions,
this value is of the order 1/sqrt(n) and everything is fine, even in
high-dimensions. For instance, this can be done if X is an orthogonal
matrix (for p=n).

The proposal made by the authors combines the ideas of the aforementioned
works with those of [Cai et al., Estimating Sparse Precision Matrix:
Optimal Rates of Convergence and Adaptive Estimation, 2012] (cf lemma 1).
Recall that Javanmard and Montanari [9] and van de Geer et al. [23]
proposed to take the ``intermediate'' estimator in the form
hat\theta^u = Lasso + M * X^T * Residual(Lasso)
where M is an estimator of the inverse covariance matrix Sigma of the
design (one should take this matrix if it is available).
An important remark made by the authors of the paper under review is
that the only property required from M in order to make the theory
work is the convergence to zero of |M*X^T*X - I|_infty. Furthermore,
the length of the CI's being proportional to [M*X^T*M^T]_{ii}, the authors
propose the smart idea of choosing M by solving a simple optimization
problem that minimizes the largest length [M*X^T*M^T]_{ii} under the
constraint that |M*X^T*X - I|_infty is bounded by a given constant.

EVALUATION
pros
- the paper reads well
- the mathematical results are correct
- the idea for choosing M is original and, in my opinion, this choice
is better than those proposed in previous works
cons
- it should be better explained why the proposed approach is better than
the simple one based on the LSE. Just saying that X is rank deficient
is not enough, since the pseudo-inverse of a rank deficient matrix may
still have nonzero entries on the diagonal.
- there are some claims in the introduction that over-evaluate the contributions
as compared to the previous work. For instance, line 53, the authors claim
that they use essentially no condition beyond the standard ones for consistency
of the Lasso. This is not true, since among the assumptions they make, I read
s_0=o(sqrt(n)/log(p)) [line 208] and norm(Sigma^{-1})_\inf = o(sqrt(n/log(p))),
which are much stronger than those required for the Lasso to be consistent.
Similarly, lines 128-129, the authors' claim that van de Geer et al. [23]
assume that n=Omega(s_0^2\log p) can be often interpreted as the results
of the paper under review do not need such a restrictive assumption (which is
not true).
- The assumption norm(Sigma^{-1})_\inf = o(sqrt(n/log(p))) seems quite strong.
Some discussion of it is necessary.
- The choice of the parameter gamma, which may be very hard since it will
depend on several unknown parameters (such as the largest diagonal
entries of Sigma and its inverse) is not discussed.
- [minor, can be easily fixed during the revision] some important
references are missing:
* there are two papers on the CI's in the sparsity setting:
- Gautier and Tsybakov [arxiv.org/pdf/1105.2454.pdf]
- Nickl and van de Geer [arxiv.org/pdf/1209.1508]
* for the scaled Lasso quoted on lines 220 and 225, the earlier work by
Belloni et al. (Biometrika 2011) should be cited.
- [minor, can be easily fixed during the revision]
Some mathematical claims lack rigor. Here is a non exhaustive list
* line 77: it should be stressed that W and X are independent
* line 182: the authors define the meaning of ``w.h.p.'', by the
convergence of the probability to zero as n\to\infty. However,
in the high-dimensional setting, n is not the only quantity that
tends to infinity. So the authors should make precise what happens
with p and s_0. Are they fix or do they go infinity with n? I guess
that the results of the paper require p to go to infinity. This
should be stated explicitly early in the text.
* line 199: I did not manage to find any clear bound on
(M_*SigmaM_*^T)_{ii} in Section 3.2.
Please be more precise in the reference.
* line 201: I do not agree that Thm 2.1 does not make any assumption on
theta_0. In fact, thm 2.1 says that X satisfies the compatibility
assumption with S the support of theta_0. If this support has more
than n elements, this may not be true. So thm 2.1 contains a
contraint on the support of theta_0.
* Eq (6) is a catastrophe from a mathematical point of view in terms
of writing. First, the left hand side is a random variable, while
the right hand side is deterministic. Second, since everything
depends on x, which varies in R, it is important to explain if o(1)
is uniform in x or not.

Summary: The paper contains an interesting improvement to the existing resuts on constructing CI's and performing hypothesis testing in high-dimensional sparse regression. The theoretical analysis is fair and interesting, but relies on some (strong) assumptions
which would require a more thorough discussion.
Author Feedback

Author rebuttal: We thank the reviewers for their detailed and thoughtful comments. Responses to the comments follow.

*REVIEWER 1:

1) Yes, you are right. We require that $s0 = o(sqrt(n)/log p)$, which is more stringent than what is required for consistency of Lasso, and this point is worth being emphasized further.

2) The diabetes example is to show how this method can be applied to real-world problems. The simulation you are suggesting are definitely useful and insightful.


*REVIEWER 4:

1) As mentioned in the introduction, the special form of the de-biased estimator resembles the one in van de Geer et al. [23]. Both the current work and [23] have the sparsity assumption s = o(\sqrt{n}/\log p) on the parameter vector.
The main step forward is the following:

In [23], the inverse covariance $\Sigma^(-1)$ is assumed to be sparse, namely with at most $s_max = o(n/log p)$ nonzero entries per row (see Theorem 2.2 in [23] and the equation above it). Needless to say this is a strong assumption that is appropriate only for some applications! By contrast,

WE DO NOT ASSUME A SPARSE INVERSE COVARIANCE (see Theorem 2.1).

Hence our method is more widely applicable. The improvement relies on a new construction of the matrix M: In [23], M is an estimator of the inverse covariance obtained by a neighborhood selection procedure. Here, we propose to choose M by solving a simple QP problem, which allows to control the bias of the estimator and also minimizes the lengths of the confidence intervals meanwhile.

Finally, we prove efficiency of our method in terms of minimax optimality (Section 3.2.1).

2) Theorem 2.1 follows readily from its non asymptotic version (Theorem A.1). Note that we are in the high-dimensional regime p>n, and therefore as the sample size n goes to infinity, p also goes to infinity. As a result the probability bound in Theorem A.1., i.e., 2(p^-c' + p^-c'') goes to zero. (choosing constant $c$ large enough, we have $c',c'' >0$). Hence, |\Delta|_\infty is smaller than $Cs0 (log p)/\sqrt(n) = o(1)$, with high probability.

3) We will add a remark stating explicitly the assumptions under which the compatibility condition for \Sigma implies a same condition for \hat{Sigma}, w.h.p.


*REVIEWER 5:

1) Regarding the simple LSE estimator you are proposing, note that the focus of the paper is on the "high-dimensional regime (p>n)". In this regime, the LSE (in general any linear estimator) is biased (see e.g. Zhang et. al [29] ). For instance, for your estimator, let X= USV^T be a (thin) SVD with U and V being respectively $nxn$, and $pxn$ matrices. Then
$\hat\theta = pinv(X^T*X)*X^T*Y = V*V^T \theta_0 +(V*S^(-1)*U^T)*w$,
where $w$ is the noise vector. The bias is then, (V*V^T - I)\theta_0. Since p>n, $V*V^T$ is a low dimensional projector and far from the identity. Hence this is an unbiased estimator and the confidence intervals as suggested are not valid.
One idea would be to remove this bias using a nonlinear estimator. Buhlmann [3] and Zhang et. al [29] propose to use ridge estimators and remove the bias using Lasso. These papers develop valid confidence intervals following this approach. However, the resulting intervals are larger by an unbounded factor \sqrt{s_0} with respect to the `noise level' $sigma*\sqrt{(\log p)/n}$. By contrast, our work achieves the ideal interval length $sigma* \sqrt{(log p)/n}$.

2) Line 53, we meant no essential assumption on the covariance matrix. Our method does not assume sparsity assumption on the covariance or the inverse covariance in contrast to previous work (see comments to Reviewer 4). But your point is fair. We will change the wording in the revision.

3) The assumption norm(Sigma^\{-1\})_\inf = o(sqrt(n/log p)) is only required in establishing minimax optimality of the procedure (Theorem 3.2). In fact it is needed to bound the variance (M*\hat(Sigma)*M^T)_ii as in Eq(17). That said, after the submission, we realized that this assumption is NOT required at all. The term ${(Sigma^(-1)* \hat(Sigma) - I) *Sigma^(-1)}_ii$, in the second line of Eq. (17), can be written as (1/n) sum_j (V_j^2 - Sigma^(-1)_ii), with V_j = (Sigma^(-1)X_j)_i. This is the average of i.i.d zero mean sub-exponential variables and using Bernstein inequality, we can show that this term is o(1), almost surely.


4) We could not discuss the choice of \gamma and \lambda due to space constraints. The choice of \lambda is to be done as for the case of simple point estimation. This has been an object of empirical and theoretical study since the invention of the Lasso and is not of course the main object of this paper.
As for the choice of \gamma, our theoretical results (Theorem A.1) imply that this is to be based only on K (which is observable), \kappa (which can be reliably estimated in many circumstances, e.g. if X has bounded entries), and C_min (which also admits consistent estimators in many contexts). Alternative data-driven approaches are also viable (e.g. testing the gaussianity of \theta^u_i claimed in Theorem A.1).

5) Other minor comments will be addressed in the revision.